# Diagnostic accuracy of serological tests for the diagnosis of Chikungunya virus infection: A systematic review and meta-analysis

**Anna Andrew**[1,2], **Tholasi Nadhan Navien**[1], **Tzi Shien Yeoh**[1], **Marimuthu Citartan**[1], **Ernest Mangantig**[1], **Magdline S. H. Sum**[3], **Ewe Seng Ch'ng**[1]*, **Thean-Hock Tang**[1]*

**1** Advanced Medical and Dental Institute, Universiti Sains Malaysia, Pulau Pinang, Malaysia, **2** Department of Paraclinical Sciences, Faculty of Medicine and Health Sciences, Universiti Malaysia Sarawak, Kota Samarahan, Sarawak, Malaysia, **3** Institute of Health and Community Medicine, Universiti Malaysia Sarawak, Kota Samarahan, Sarawak, Malaysia

* eschng@usm.my (ESC); tangth@usm.my (THT)

**Data Availability Statement:** All relevant data are within the manuscript and its Supporting Information files.

## Abstract

### Background

Chikungunya virus (CHIKV) causes febrile illnesses and has always been misdiagnosed as other viral infections, such as dengue and Zika; thus, a laboratory test is needed. Serological tests are commonly used to diagnose CHIKV infection, but their accuracy is questionable due to varying degrees of reported sensitivities and specificities. Herein, we conducted a systematic review and meta-analysis to evaluate the diagnostic accuracy of serological tests currently available for CHIKV.

### Methodology and principal findings

A literature search was performed in PubMed, CINAHL Complete, and Scopus databases from the 1st December 2020 until 22nd April 2021. Studies reporting sensitivity and specificity of serological tests against CHIKV that used whole blood, serum, or plasma were included. QUADAS-2 tool was used to assess the risk of bias and applicability, while R software was used for statistical analyses.

Thirty-five studies were included in this meta-analysis; 72 index test data were extracted and analysed. Rapid and ELISA-based antigen tests had a pooled sensitivity of 85.8% and 82.2%, respectively, and a pooled specificity of 96.1% and 96.0%, respectively. According to our meta-analysis, antigen detection tests serve as a good diagnostic test for acute-phase samples. The IgM detection tests had more than 90% diagnostic accuracy for ELISA-based tests, immunofluorescence assays, in-house developed tests, and samples collected after seven days of symptom onset. Conversely, low sensitivity was found for the IgM rapid test (42.3%), commercial test (78.6%), and for samples collected less than seven of symptom onset (26.2%). Although IgM antibodies start to develop on day 2 of CHIKV infection, our meta-analysis revealed that the IgM detection test is not recommended for acute-phase samples. The diagnostic performance of the IgG detection tests was more than 93% regardless of the test formats and whether the test was commercially available or developed in-

**Funding:** Postgraduate candidacy of A. A. and T.N.
N. were supported by government scholarships:
Skim Latihan Akademik Bumiputera (SLAB) from
the Government of Malaysia and University Sains
Malaysia (USM) fellowship (IPS/Fellowship2019/
IPG), respectively. The funders had no role in study
design, data collection and analysis, decision to
publish, or preparation of the manuscript.

**Competing interests:** The authors have declared
that no competing interests exist.

house. The use of samples collected after seven days of symptom onset for the IgG detection test suggests that IgG antibodies can be detected in the convalescent-phase samples. Additionally, we evaluated commercial IgM and IgG tests for CHIKV and found that ELISA-based and IFA commercial tests manufactured by Euroimmun (Lübeck, Germany), Abcam (Cambridge, UK), and Inbios (Seattle, WA) had diagnostic accuracy of above 90%, which was similar to the manufacturers' claim.

## Conclusion

Based on our meta-analysis, antigen or antibody-based serological tests can be used to diagnose CHIKV reliably, depending on the time of sample collection. The antigen detection tests serve as a good diagnostic test for samples collected during the acute phase ($\leq$7 days post symptom onset) of CHIKV infection. Likewise, IgM and IgG detection tests can be used for samples collected in the convalescent phase (>7 days post symptom onset). In correlation to the clinical presentation of the patients, the combination of the IgM and IgG tests can differentiate recent and past infections.

## Author summary

Chikungunya virus (CHIKV) causes non-specific symptoms such as fever, and the infection is sometimes misinterpreted as other viral infections, such as dengue and Zika. Although serological tests are commonly used to diagnose CHIKV infection, the reliability of these tests is questionable due to their highly variable performance. A systematic review and meta-analysis were performed to determine the diagnostic accuracy of these serological tests. As the analytes (antigen and antibodies) are present in the patient's sample at different time points of CHIKV infection, we analysed the diagnostic performance of serological tests detecting CHIKV antigen, IgM, and IgG antibodies. Our meta-analysis showed that antigen or antibody-based serological tests could reliably be used to diagnose CHIKV, depending on the time of sample collection. Antigen detection test serves as a good diagnostic test for samples collected within the acute phase (1 to 7 days) of CHIKV infections. On the other hand, the IgM and IgG tests can be used for convalescent-phase (>7 days of symptom onset) samples, differentiating recent and past CHIKV infections. Although IgM antibodies start to develop as early as 2 to 4 days of CHIKV infection, our result showed that the IgM detection tests for acute-phase samples exhibited low accuracy. Thus, the IgM detection test is not recommended for samples collected <7 days of symptoms onset.

## 1. Introduction

Chikungunya virus (CHIKV) is transmitted to humans by Aedes mosquito bite. First isolated in Tanzania in 1953 [1], CHIKV was restricted to sporadic outbreaks in Africa and Asia. The three genotypes of CHIKV are designated after its geographical origins: East/Central/South/African (ECSA), West African, and Asian [2]. A genotypic shift of the CHIKV from Asian to ECSA was observed during the massive Indian Ocean outbreak in 2004, affecting millions of people [3]. ECSA genotype of CHIKV then continues to cause outbreaks in India and other parts of Asia [4,5]. Due to increased human movement and virus adaptability inside vectors,

CHIKV has been recorded in nonendemic regions of the world [6,7]. To date, CHIKV is widespread in the Americas, Asia, and Africa [8], and the risk of reemergence and transmission remains a public health concern.

Chikungunya fever is caused by CHIKV and is characterised by fever, rashes, and severe joint pain. The symptoms can progress to chronic joint pain, affecting the patient's quality of life [9]. Since no licensed vaccines or therapies are available yet against CHIKV, early diagnosis may allow for early control strategies, preventing further outbreaks. As the clinical symptoms of CHIKV infections are similar to other viral illnesses, a reliable, sensitive, and specific laboratory test that can distinguish CHIKV infections from other viral infections is urgently needed.

According to World Health Organization (WHO) guidelines, the three main laboratory tests for diagnosing CHIKV infections are virus isolation, serological tests, and molecular technique of polymerase chain reaction (PCR) [10]. The choice of tests depends on the number of days from the symptom onset. Virus isolation and quantitative reverse transcription-PCR (qRT-PCR) are recommended for samples collected within the first five days of illness. Meanwhile, serology tests are used for samples collected 5 days after the onset of illness. According to WHO, the Immunoglobulin M (IgM) ELISA is the most prevalent serology test used to diagnose CHIKV infection.

Compared to the standard methods such as virus isolation, qRT-PCR, and plaque reduction neutralisation tests (PRNT), antigen and antibody-based serological tests are easier to perform, cost-effective, and require minimum resources. Following the outbreaks in the Indian Ocean in 2004, studies on CHIKV serological tests increased tremendously [11]. However, the diagnostic accuracy of these serological tests is unknown due to various degrees of reported sensitivities and specificities. To assess the diagnostic accuracy of the existing CHIKV serological assays, we performed a systematic review and meta-analysis. As different analytes were detected at different time points of sample collection, the diagnostic performance of serological tests identifying CHIKV antigen, IgM and IgG antibodies was determined.

## 2. Methods

### 2.1 Study registration

We adopted the preferred reporting items for a systematic review and meta-analysis of diagnostic test accuracy (PRISMA-DTA) guideline in preparing this report [12]. This systematic review was registered in the PROSPERO database under CRD42021227523.

### 2.2 Inclusion and exclusion criteria

Inclusion criteria in this systematic review were studies that 1) used suspected chikungunya patients regardless of age, gender, or other health status; 2) assessed the diagnostic performance of either antigen or antibody-based serological tests; 3) used either virus isolation, cell culture, or molecular methods as the reference standard for antigen detection test; 4) used either human serum, plasma, or whole blood as the samples; 5) contained sufficient information to tabulate 2 x 2 contingency table. Other research materials such as conference abstracts, commentaries, review articles, editorials, notes, and studies that did not specify the reference methods were excluded.

### 2.3 Literature search strategy

The literature search was performed in PubMed, CINAHL Complete, and Scopus databases from the 1$^{st}$ December 2020 until 22$^{nd}$ April 2021. The search was limited to journal articles written in English and published from the year 2000 onwards. The year 2000 was chosen as the

cutoff year because CHIKV infection had been neglected before the unprecedented magnitude outbreak in Indian Ocean territories in 2004 [11]. Therefore, not many studies on CHIKV serological tests were available before the year 2000. We also screened through the reference lists of all the included studies to identify the relevant literature. The detailed search strategies for each database are shown in the S1 Appendix. All the articles were imported into Endnote X9.2 (Clarivate Analytics, USA) for the study selection. After the full-text screening stage, we documented the reasons for studies excluded in a PRISMA flow diagram.

## 2.4 Data extraction

According to the inclusion criteria mentioned above, the data extraction was done independently by two reviewers (AA and YTS). Other than the true positive, false positive, false negative, and true negative, information such as author information, study design, sample size, index test format, reference test description, and the time of sample collection were extracted from these articles. Any ambiguities of the extracted data were resolved by mutual agreement between authors.

A study can evaluate more than one index test, and all the index tests data reported in each study were extracted. One of the studies [13] reported diagnostic accuracy from three different laboratories, namely CDC, CARPHA, and NML. As each of these laboratories evaluated a different set of index tests, we named these studies according to the laboratories (i.e., Johnson (CDC), Johnson (CARPHA), and Johnson (NML)). For studies developing serological tests either with different antigens or antibodies of the same test format, only the optimised index test data (highest diagnostic accuracy) were extracted for analysis.

## 2.5 Quality assessment

**2.5.1 Study design.** Analysis based on study design was done to determine each study's reliability and quality of evidence. We divided the study design into the cohort, case-control, and partial cohort partial case-control study. The cohort study was a study that used suspected chikungunya patient (patient presented with fever and/or rash, myalgia, or arthralgia) samples to determine the accuracy of a test. The case-control study was a study that used confirmed chikungunya positive patient samples to determine the test sensitivity and serum samples from healthy individuals to determine the test specificity. The partial cohort and partial case-control study, on the other hand, assessed the diagnostic accuracy of the test using cohort samples as well as other pathogen positive samples (for example, dengue, Ross River virus (RRV), O'nyong-nyong virus (ONN)). For this analysis, the cohort was pooled together with the partial cohort partial case-control study design and compared with the case-control study design.

**2.5.2 Risk of bias and applicability.** The Quality Assessment of Diagnostic Accuracy Studies (QUADAS-2) tool was used to evaluate the quality and bias of each study [14]. The four domains evaluated were patient selection, index test, reference standard, and flow and timing (flow of patients through the study and timing of the index test and reference standard). The risk of bias was described as either low, high, or unclear in each domain, while concerns regarding the applicability were only assessed for the first three domains. Slight modifications were done to the signalling questions from the original tool. When more than one signalling question in a domain answered "no" or "unclear", that domain will be rated as a high risk of bias (see S2 Appendix). Two reviewers (AA and NTN) independently assessed the quality of each study, and any disagreements were resolved through a consensual approach. The graph for the risk of bias and the applicability concern was generated using Review Manager 5.4 software.

### 2.6 Data analysis

A meta-analysis was performed in R software version 4.0.5 using the "meta" package. Pooled estimates of sensitivity (the probability of a test to identify those with the disease correctly) and specificity (the probability of a test to exclude those without disease correctly) with 95% confidence intervals were calculated using a random-effect model (Maximum-likelihood estimation), and the summary was presented in a paired forest plot. A random-effect model was chosen to consider the heterogeneity present within and between the studies [15]. Heterogeneity between studies was estimated using $I^2$ statistics (total variation across the studies). The $I^2$ value of 75% and above was rated as high, 50–74% as medium, and 49–25% as low heterogeneity. A funnel plot asymmetry test was used to assess publication bias [16].

**2.6.1 Subgroup analysis.**   The source of heterogeneity was investigated by stratifying the data based on analytes detected by the serological tests, namely antigen, IgM, and IgG antibodies. We further assessed the source of heterogeneity by classifying the data based on test formats (ELISA-based, IFA, and rapid test), commercial versus in-house developed test, and time of samples collection (samples collected day 1 to 7 and after 7 days from the onset of clinical symptoms).

For commercial tests (specific brand) with two or more diagnostic accuracy studies, meta-analyses were done according to the individual commercial kit. We included only samples collected after 7 days from the onset of clinical symptoms for this analysis. The commercial kit sensitivity and specificity reported by the manufacturers were also compared with the accuracy reported in this study. All the analyses were done using R software to calculate pooled estimates of sensitivity and specificity. The Mann-Whitney or Kruskal-Wallis test was used to compare the sensitivity and specificity values between groups.

## 3. Results

### 3.1 Literature search results

A total of 563 articles were identified through the mentioned databases. After removing duplicates, the remaining articles underwent title and abstract screening. Thereafter, a total of 40 articles were subjected to inclusion criteria evaluation. Three studies did not specify the reference standard [17–19], while one did not provide sufficient details for constructing the 2 x 2 contingency table [20]. In addition, one particular study involving cerebrospinal fluid (CSF) samples was excluded [21]. Finally, the remaining thirty-five articles were subjected to full-text reviewing for meta-analysis (Fig 1).

### 3.2 Characteristics of the included studies

We tabulated 72 sets of data from the 35 studies. Of the 72 tests assessed, 7 were antigen detection tests, 48 were IgM, 15 were IgG, and two were neutralising antibodies detection tests. Tables 1–4 show the data for each analyte, and Table 5 shows the summary characteristics of the studies that were included. A total of 10563 participants were included in this study, with 880 participants tested for antigen, 7613 participants for IgM, 1539 participants for IgG, and 531 participants for neutralising antibodies. Most of the studies (70%) did not specify the time of sample collection and the clinical background of the study participants. Only five studies (14.3%) specified that the samples were collected from hospitalised patients, and six studies (17.1%) used patient samples collected during CHIKV outbreaks.

### 3.3 Diagnostic accuracy of serological tests for CHIKV infection

A meta-analysis based on the analytes (CHIKV antigen, IgM, and IgG antibodies) was done in this study. Forest plot for antigen, IgM, IgG, and neutralising antibodies (See S1 Fig) shows

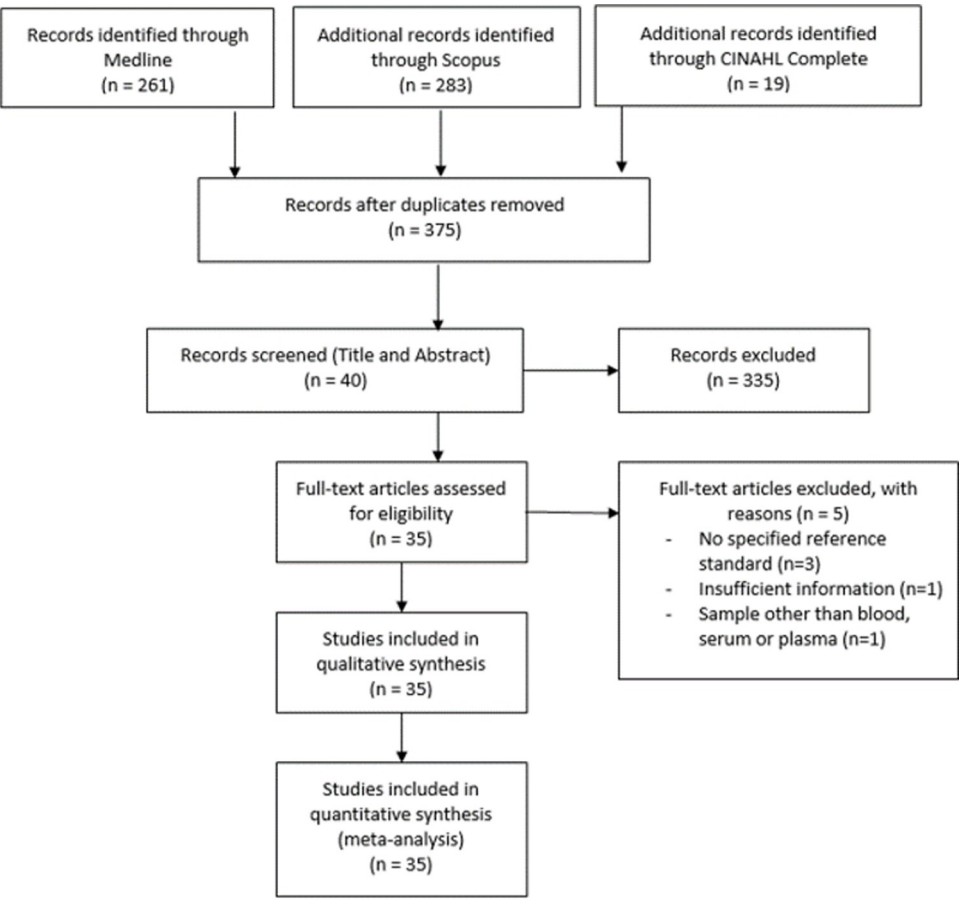

**Fig 1. PRISMA flow diagram.**

that the sensitivity across studies ranged from 0 to 1.0, while the specificity ranged from 0.73 to 1.0. Following the available information, the source of heterogeneity was further evaluated based on the test format, in-house developed versus commercial test, and time of sample collection. As there were only two studies on neutralising antibodies detection tests, subgroup analysis was not performed.

### 3.4 Antigen detection test

All seven antigen detection studies used molecular method and/or virus isolation as the reference test, and none of the antigen detection tests was commercially available. The samples used for antigen detection test were acute samples ranging from 1 to 20 days post symptom onset (Table 1). The forest plot for antigen detection test based on test format is shown in Fig 2. Meta-analysis showed no difference in the diagnostic performance between rapid and ELISA-based tests ($P = >0.05$) (Table 6). The heterogeneity for the sensitivity was high for both test formats, while moderate heterogeneity was observed for the specificity of the rapid antigen detection test.

### 3.5 IgM detection test

A variety of reference standards were used in the diagnostic accuracy studies of the IgM detection test, which included molecular methods, in-house developed serology tests, and

**Table 1. Characteristics of the studies on antigen detection tests included in the meta-analysis.**

| Author | Year | Study design | Reference test | Index test format | Index test (Commercial/ In-house) | Time of sample collection (day of post symptom onset) | Total number of samples | TP | FP | FN | TN | Ref |
|---|---|---|---|---|---|---|---|---|---|---|---|---|
| Huits | 2018 | Partial cohort and case-control | RT-PCR | Rapid test | In-house | 1 to 10 | 97 | 18 | 12 | 21 | 46 | [22] |
| Jain | 2018 | Case-control | qRT-PCR | Rapid test | In-house | 1 to 15 | 123 | 74 | 2 | 5 | 42 | [23] |
| Kashyap | 2010 | Cohort | RT-PCR or qRT-PCR or virus isolation | Antigen Indirect ELISA | In-house | 1 to >20 | 128 | 98 | 2 | 11 | 17 | [24] |
| Khan | 2014 | Cohort | RT-PCR | Antigen capture ELISA | In-house | NA | 60 | 35 | 0 | 3 | 22 | [25] |
| Okabayashi | 2015 | Cohort | RT-PCR | Rapid test | In-house | NA | 112 | 68 | 2 | 8 | 34 | [26] |
| Reddy | 2020 | Cohort | qRT-PCR | Antigen Indirect ELISA | In-house | 1 to 5 | 160 | 51 | 2 | 49 | 58 | [27] |
| Suzuki | 2020 | Partial cohort and case-control | RT-PCR | Rapid test | In-house | 1 to 7 | 200 | 92 | 0 | 8 | 100 | [28] |

Note: TP, true positive; FP, false positive; FN, false negative; TN, true negative; Ref, reference; NA, not available

commercial kits (Table 2). Some studies used the molecular method to confirm CHIKV infection for samples collected on the first day of symptoms appeared, then later samples from the same patients were collected for the IgM detection test.

Subgroup analyses were conducted for the IgM detection test based on test format, in-house developed versus commercial, and sampling time. The three test formats available for IgM detection tests were rapid, ELISA-based, and immunofluorescence assay (IFA). Regardless of the test formats, the forest plot (Fig 3) shows that the sensitivity estimates vary more widely than the specificity estimates. Meanwhile, meta-analyses revealed that the rapid tests had the poorest sensitivity, 42.3% (95% CI 19.2 to 69.4) (Table 7). The sensitivity of the rapid tests (42.3%; 95% CI 19.2 to 69.4) was statistically different from ELISA-based (93.4%; 95% CI 81.7 to 97.8; $P = 0.002$) and IFA (99.3%; 95% CI 69.4 to 100; $P = 0.027$), while no significant difference was found in the sensitivity of IFA and ELISA-based tests ($P = 0.414$).

More than half of the IgM detection tests investigated (60%) were commercially available, and the sensitivity of these tests was highly variable compared to the in-house developed test (Fig 4). According to our meta-analysis, the diagnostic accuracy of in-house developed tests was significantly higher than commercial IgM tests (Table 7).

The sample collection time for the IgM detection tests ranges from day 1 to day 40 after the onset of symptoms. For studies that provide sample collection time, we categorised sample collected ≤ 7 days post symptom onset as acute-phase samples and >7 days post symptom onset as convalescent-phase samples (Table 2). The forest plot (Fig 5) shows that the sensitivity estimates for samples collected ≤ 7 days of symptoms onset mostly lies on the left side of the plot. Consistent with this observation, our meta-analysis shows that the sensitivity for the samples collected ≤ 7 days of symptoms onset was significantly lower than samples collected >7 days post symptom onset (Table 7). These results indicate that the IgM detection test had low accuracy for acute-phase samples.

The sensitivity heterogeneity was moderate to high (73.7 to 96.5%) across all subgroup studies for IgM detection tests. In comparison, the test specificity showed low to moderate (0 to 72.0%) heterogeneity (Table 7).

**Table 2. Characteristics of studies on IgM detection tests included in the meta-analysis.**

| Author | Year | Study design | Reference test | Index test format | Index test (Commercial/ In-house) | Time of sample collection (day of post symptom onset) | Total number of samples | TP | FP | FN | TN | Ref |
|---|---|---|---|---|---|---|---|---|---|---|---|---|
| Bagno | 2020 | Partial cohort and case-control | Anti-chikungunya IgG ELISA kit (Euroimmun, Germany) | IgM Indirect ELISA | In-house | NA | 144 | 57 | 1 | 10 | 76 | [29] |
| Bhatnagar | 2015 | Case-control | RT-PCR and IgM kit | IgM Indirect ELISA | In-house | 7 to 23 [b] | 90 | 45 | 0 | 0 | 45 | [30] |
| Blacksell | 2011 | Cohort | Hemagglutination inhibition (HI) and/or IgM antibody capture ELISA and/or RT-PCR | Rapid test | Commercial (SD Diagnostics) | 3 to 7 [a] | 292 | 2 | 15 | 50 | 225 | [31] |
| Blacksell | 2011 | Cohort | Hemagglutination inhibition (HI) and/or IgM antibody capture ELISA and/or RT-PCR | MAC-ELISA | Commercial (SD Diagnostics) | 3 to 7 [a] | 292 | 2 | 18 | 50 | 222 | [31] |
| Blacksell | 2011 | Cohort | Hemagglutination inhibition (HI) and/or IgM antibody capture ELISA and/or RT-PCR | MAC-ELISA | Commercial (SD Diagnostics) | 19 to 30 [b] | 292 | 44 | 21 | 8 | 219 | [31] |
| Cho | 2008 | Case-control | IgM capture ELISA (Lyon, France) | IgM Indirect ELISA (E1) | In-house | NA | 60 | 31 | 0 | 9 | 20 | [32] |
| Cho | 2008 | Case-control | IgM capture ELISA (Lyon, France) | IgM Indirect ELISA (E2) | In-house | NA | 60 | 36 | 0 | 4 | 20 | [32] |
| Cho | 2008 | Case-control | IgM capture ELISA (Lyon, France) | IgM Indirect ELISA (Capsid) | In-house | NA | 60 | 34 | 0 | 6 | 20 | [33] |
| Cho | 2008 | Case-control | IgM capture ELISA (Lyon, France) | Rapid test (Capsid) | In-house | NA | 60 | 35 | 0 | 5 | 20 | [33] |
| Damle | 2016 | Cohort | MAC-ELISA (National Institute of Virology, Pune) | MAC-ELISA (Capsid) | In-house | NA | 248 | 67 | 0 | 10 | 171 | [34] |
| Galo | 2017 | Cohort | CDC-MAC-ELISA (Atlanta, Georgia, United States) | MAC-ELISA | In-house | ~5.9 [a] | 198 | 113 | 1 | 3 | 81 | [35] |
| Johnson (CDC) | 2016 | Partial cohort and case-control | CDC MAC-ELISA and PRNT | IgM Indirect ELISA | Commercial (Euroimmun) | 2 to 33 | 92 | 51 | 1 | 1 | 39 | [13] |
| Johnson (CDC) | 2016 | Partial cohort and case-control | CDC MAC-ELISA and PRNT | IFA | Commercial (Euroimmun) | 2 to 33 | 75 | 34 | 3 | 0 | 38 | [13] |
| Johnson (CDC) | 2016 | Partial cohort and case-control | CDC MAC-ELISA and PRNT | MAC-ELISA | Commercial (Abcam) | 2 to 33 | 70 | 36 | 1 | 0 | 33 | [13] |
| Johnson (CDC) | 2016 | Partial cohort and case-control | CDC MAC-ELISA and PRNT | MAC-ELISA | Commercial (InBios) | 2 to 33 | 71 | 36 | 0 | 0 | 35 | [13] |
| Johnson (CDC) | 2016 | Partial cohort and case-control | CDC MAC-ELISA and PRNT | MAC-ELISA | Commercial (CTK Biotech) | 2 to 33 | 20 | 2 | 0 | 14 | 4 | [13] |
| Johnson (CDC) | 2016 | Partial cohort and case-control | CDC MAC-ELISA and PRNT | MAC-ELISA | Commercial (Genway) | 2 to 33 | 43 | 0 | 0 | 27 | 16 | [13] |

*(Continued)*

**Table 2.** (*Continued*)

| Author | Year | Study design | Reference test | Index test format | Index test (Commercial/ In-house) | Time of sample collection (day of post symptom onset) | Total number of samples | TP | FP | FN | TN | Ref |
|---|---|---|---|---|---|---|---|---|---|---|---|---|
| Johnson (CDC) | 2016 | Partial cohort and case-control | CDC MAC-ELISA and PRNT | MAC-ELISA | Commercial (SD Diagnostics) | 2 to 33 | 44 | 12 | 2 | 19 | 11 | [13] |
| Johnson (CDC) | 2016 | Partial cohort and case-control | CDC MAC-ELISA and PRNT | Rapid test | Commercial (SD Diagnostics) | 2 to 33 | 31 | 0 | 0 | 24 | 7 | [13] |
| Johnson (CDC) | 2016 | Partial cohort and case-control | CDC MAC-ELISA and PRNT | Rapid test | Commercial (CTK Biotech) | 2 to 33 | 27 | 3 | 0 | 20 | 4 | [13] |
| Johnson (CARPHA) | 2016 | Partial cohort and case-control | CDC MAC-ELISA and PRNT | Indirect ELISA | Commercial (Euroimmun) | NA | 36 | 26 | 0 | 0 | 10 | [13] |
| Johnson (CARPHA) | 2016 | Partial cohort and case-control | CDC MAC-ELISA and PRNT | IFA | Commercial (Euroimmun) | NA | 33 | 21 | 1 | 0 | 11 | [13] |
| Johnson (CARPHA) | 2016 | Partial cohort and case-control | CDC MAC-ELISA and PRNT | MAC-ELISA | Commercial (Abcam) | NA | 46 | 36 | 0 | 0 | 10 | [13] |
| Johnson (CARPHA) | 2016 | Partial cohort and case-control | CDC MAC-ELISA and PRNT | MAC-ELISA | Commercial (InBios) | NA | 41 | 27 | 1 | 0 | 13 | [13] |
| Johnson (NML) | 2016 | Partial cohort and case-control | CDC MAC-ELISA and PRNT and/or qRT-PCR and/or hemagglutination inhibition assay | Indirect ELISA | Commercial (Euroimmun) | NA | 247 | 94 | 6 | 6 | 141 | [13] |
| Khan | 2014 | Cohort | RT-PCR and in-house indirect IgM ELISA | Indirect ELISA | In-house | NA | 96 | 68 | 2 | 0 | 26 | [25] |
| Khan | 2014 | Cohort | RT-PCR and in-house indirect IgM ELISA | MAC-ELISA | In-house | NA | 96 | 67 | 0 | 1 | 28 | [25] |
| Kikuti | 2020 | Cohort | RT-PCR | MAC-ELISA | Commercial (InBios) | 1 to 7 [a] | 369 | 6 | 5 | 144 | 214 | [36] |
| Kikuti | 2020 | Cohort | RT-PCR | MAC-ELISA | Commercial (InBios) | 8 to >30 [b] | 266 | 61 | 19 | 5 | 181 | [36] |
| Kikuti | 2020 | Cohort | RT-PCR | Indirect ELISA | Commercial (Euroimmun) | 1 to 7 [a] | 354 | 15 | 24 | 130 | 185 | [36] |
| Kikuti | 2020 | Cohort | RT-PCR | Indirect ELISA | Commercial (Euroimmun) | 8 to >30 [b] | 258 | 63 | 31 | 2 | 162 | [36] |
| Kosasih | 2012 | Partial cohort and case-control | In-house IgM capture ELISA and/or RT-PCR | Rapid test | Commercial (CTK Biotech) | 1 to ≥21 | 206 | 27 | 0 | 105 | 74 | [37] |
| Kosasih | 2012 | Partial cohort and case-control | In-house IgM capture ELISA and/or RT-PCR | Rapid test | Commercial (SD Diagnostics) | 1 to ≥21 | 206 | 67 | 8 | 65 | 66 | [37] |

(*Continued*)

**Table 2.** (Continued)

| Author | Year | Study design | Reference test | Index test format | Index test (Commercial/ In-house) | Time of sample collection (day of post symptom onset) | Total number of samples | TP | FP | FN | TN | Ref |
|---|---|---|---|---|---|---|---|---|---|---|---|---|
| Lee | 2020 | Case-control | Euroimmun and Inbios IgM ELISA | Rapid test | Commercial (Boditech Med Inc) | NA | 220 | 57 | 1 | 0 | 162 | [38] |
| Litzba | 2008 | Case-control | In-house IgM capture ELISA or in-house IIFT | IFA | Commercial (Euroimmun) | NA | 246 | 127 | 2 | 4 | 113 | [39] |
| Matheus | 2015 | Cohort | qRT-PCR and/or MAC-ELISA | MAC-ELISA | In-house | >5 [b] | 58 | 15 | 1 | 0 | 42 | [40] |
| Mendoza | 2019 | Case-control | Plaque reduction neutralization test (PRNT) and/or RT-PCR | IgM Indirect ELISA | Commercial (Euroimmun) | NA | 212 | 154 | 0 | 7 | 51 | [41] |
| Prat | 2014 | Partial cohort and case-control | In-house MAC-ELISA and PRNT | Rapid test | Commercial (SD Diagnostics) | NA | 25 | 3 | 4 | 7 | 11 | [42] |
| Prat | 2014 | Partial cohort and case-control | In-house MAC-ELISA and PRNT | Rapid test | Commercial (CTK Biotech) | NA | 25 | 2 | 1 | 8 | 14 | [42] |
| Prat | 2014 | Partial cohort and case-control | In-house MAC-ELISA and PRNT | MAC-ELISA | Commercial (IBL International) | NA | 53 | 22 | 3 | 6 | 22 | [42] |
| Prat | 2014 | Partial cohort and case-control | In-house MAC-ELISA and PRNT | IgM Indirect ELISA | Commercial (Euroimmun) | NA | 50 | 22 | 5 | 4 | 19 | [42] |
| Priya | 2014 | Partial cohort and case-control | SD IgM ELISA (Standard Diagnostics, South Korea) | IgM Indirect ELISA | In-house | 3 to 10 [b] | 90 | 48 | 2 | 0 | 40 | [43] |
| Rianthavorn | 2010 | Cohort | Semi-nested RT-PCR and ELISA kit (SD BIOLINE) | Rapid test | Commercial (SD Diagnostics) | 1 to 6 [a] | 367 | 33 | 17 | 153 | 164 | [44] |
| Rianthavorn | 2010 | Cohort | Semi-nested RT-PCR and ELISA kit (SD BIOLINE) | Rapid test | Commercial (SD Diagnostics) | 7 to >14 [b] | 160 | 67 | 23 | 14 | 56 | [44] |
| Theillet | 2019 | Case-control | In-house MAC-ELISA | Rapid test | In-house | NA | 78 | 24 | 1 | 10 | 43 | [45] |
| Verma | 2014 | Case-control | RT-PCR or IgM kit | IgM Indirect ELISA | In-house | 7 to 15 [b] | 195 | 115 | 0 | 8 | 72 | [46] |
| Wang | 2019 | Partial cohort and case-control | ELISA kit (Euroimmun) | Rapid test | In-house | NA | 109 | 10 | 3 | 2 | 94 | [47] |
| Wasonga | 2015 | Cohort | IgM-capture ELISA (CDC) and focus reduction neutralization test | MAC-ELISA | In-house | NA | 148 | 51 | 3 | 5 | 89 | [48] |
| Yap | 2010 | Partial cohort and case-control | RT-PCR and IgM serology | Rapid test | Commercial (CTK Biotech) | 1 to 6 [a] | 141 | 24 | 0 | 67 | 50 | [49] |
| Yap | 2010 | Partial cohort and case-control | RT-PCR and IgM serology | Rapid test | Commercial (CTK Biotech) | 7 to 40 [b] | 93 | 23 | 0 | 20 | 50 | [49] |

(*Continued*)

**Table 2.** (Continued)

| Author | Year | Study design | Reference test | Index test format | Index test (Commercial/ In-house) | Time of sample collection (day of post symptom onset) | Total number of samples | TP | FP | FN | TN | Ref |
|---|---|---|---|---|---|---|---|---|---|---|---|---|
| Yap | 2010 | Partial cohort and case-control | RT-PCR and IgM serology | IFA | Commercial (Euroimmun) | 1 to 6 [a] | 240 | 92 | 0 | 98 | 50 | [49] |
| Yap | 2010 | Partial cohort and case-control | RT-PCR and IgM serology | IFA | Commercial (Euroimmun) | 7 to 40 [b] | 145 | 95 | 0 | 0 | 50 | [49] |
| Yap | 2010 | Partial cohort and case-control | RT-PCR and IgM serology | MAC-ELISA (226A) | In-house | 1 to 6 [a] | 240 | 96 | 2 | 94 | 48 | [49] |
| Yap | 2010 | Partial cohort and case-control | RT-PCR and IgM serology | MAC-ELISA (226A) | In-house | 7 to 40 [b] | 145 | 95 | 2 | 0 | 48 | [49] |
| Yap | 2010 | Partial cohort and case-control | RT-PCR and IgM serology | MAC-ELISA (226V) | In-house | 1 to 6 [a] | 240 | 118 | 2 | 72 | 48 | [49] |
| Yap | 2010 | Partial cohort and case-control | RT-PCR and IgM serology | MAC-ELISA (226V) | In-house | 7 to 40 [b] | 145 | 95 | 2 | 0 | 48 | [49] |

Note: TP, true positive; FP, false positive; FN, false negative; TN, true negative; Ref, reference; NA, not available

[a] Acute samples

[b] Convalescent samples

## 3.6 IgG detection test

The reference standards used for IgG detection test studies include the commercial kits, in-house developed ELISA, IFA, or PRNT. The time of sample collection for IgG detection tests ranges from 7 to 90 days of post symptom onset.

Subgroup analysis based on test format and in-house developed versus commercial tests were done for the IgG detection test. The forest plot for the three different test formats (ELISA-based, rapid test, and IFA) was shown in Fig 6. We found no difference ($P = >0.05$) in the diagnostic performance of the three different test formats (IFA, ELISA-based and rapid test), and rapid tests showed the highest accuracy (Table 8). Although there was no difference, the IFA and rapid test accuracy have to be interpreted with caution as the sample size for IFA and the rapid IgG detection test was relatively low compared to the ELISA-based test.

We compared the diagnostic performance of commercial and in-house developed IgG tests. Fig 7 shows the forest plot for commercial and in-house developed IgG tests, and our analysis showed no difference in the diagnostic accuracy of the two tests (Table 8). In summary, the CHIKV IgG detection tests had a high diagnostic accuracy with more than 93% sensitivity and specificity regardless of the test format and commercial or in-house developed test.

The sensitivity heterogeneity in the subgroup analysis for IgG tests ranged from medium to high ($I^2$ of 72.4 to 83.6) except for the IFA and rapid test, which showed no heterogeneity (Table 8). There was no heterogeneity in the specificity of all the IgG detection tests.

### 3.7 Subgroup analysis of commercial serological tests for CHIKV

A meta-analysis was performed for nine commercial tests detecting IgM and IgG antibodies (Table 9). The data for meta-analysis based on test format were available in sections A-C of S1 Table. Most commercial kits indicated testing using samples taken between 6 and 8 days after symptom onset. Therefore, data from samples collected less than 7 days after symptom onset were eliminated from the analysis. The commercial kit studies mostly use cohort or partial cohort partial case-control study designs. Case-control study design was used in only two of the studies [39,41].

**Table 3. Characteristics of studies on IgG detection tests included in the meta-analysis.**

| Author | Year | Study design | Reference test | Index test format | Index test (Commercial/ In-house) | Time of sample collection (day of post symptom onset) | Total number of samples | TP | FP | FN | TN | Ref |
|---|---|---|---|---|---|---|---|---|---|---|---|---|
| Bagno | 2020 | Partial cohort and case-control | Anti-chikungunya IgG ELISA kit (Euroimmun, Germany) | IgG Indirect ELISA | In-house | NA | 156 | 69 | 3 | 1 | 83 | [29] |
| De Salazar | 2017 | Partial cohort and case-control | In-house ELISA (CDC, Atlanta, United States) | GAC-ELISA | Commercial (InBios) | 15 to 90 | 36 | 13 | 2 | 1 | 20 | [50] |
| De Salazar | 2017 | Partial cohort and case-control | In-house ELISA (CDC, Atlanta, United States) | IgG Indirect ELISA | Commercial (Euroimmun) | 15 to 90 | 36 | 14 | 4 | 0 | 18 | [50] |
| De Salazar | 2017 | Partial cohort and case-control | In-house ELISA (CDC, Atlanta, United States) | IFA | Commercial (Euroimmun) | 15 to 90 | 36 | 14 | 2 | 0 | 20 | [50] |
| Fumagalli | 2018 | Cohort | Plaque reduction neutralization test | IgG Indirect ELISA | In-house | NA | 59 | 26 | 0 | 3 | 30 | [51] |
| Kowalzik | 2008 | Case-control | IFA | Rapid test | In-house | NA | 130 | 22 | 0 | 8 | 100 | [52] |
| Kumar | 2014 | Partial cohort and case-control | IgG IFA (Euroimmun) | GAC-ELISA | In-house | ≥ 9 | 141 | 83 | 5 | 17 | 36 | [53] |
| Lee | 2020 | Case-control | Euroimmun and Inbios IgG ELISA | Rapid test | Commercial (Boditech Med Inc) | NA | 199 | 36 | 0 | 0 | 163 | [38] |
| Litzba | 2008 | Case-control | Indirect IgG ELISA or In-house IIFT | IFA | Commercial (Euroimmun) | NA | 207 | 83 | 0 | 4 | 120 | [39] |
| Mendoza | 2019 | Case-control | Plaque reduction neutralisation test and/or RT-PCR | IgG Indirect ELISA | Commercial (Euroimmun) | NA | 212 | 155 | 1 | 6 | 50 | [41] |
| Mendoza | 2019 | Case-control | Plaque reduction neutralisation test and/or RT-PCR | GAC-ELISA | Commercial (Abcam) | NA | 212 | 155 | 0 | 6 | 51 | [41] |
| Prat | 2014 | Partial cohort and case-control | In-house ELISA and PRNT | GAC-ELISA | Commercial (IBL International) | NA | 53 | 15 | 1 | 13 | 24 | [42] |
| Prat | 2014 | Partial cohort and case-control | In-house ELISA and PRNT | IgG Indirect ELISA | Commercial (Euroimmun) | NA | 47 | 22 | 3 | 3 | 19 | [42] |
| Verma | 2014 | Case-control | RT-PCR or IgM kit | IgG Indirect ELISA | In-house | 7 to 15 | 195 | 117 | 0 | 6 | 72 | [46] |
| Wang | 2019 | Partial cohort and case-control | ELISA kit (Euroimmun) | Rapid test | In-house | NA | 109 | 29 | 0 | 0 | 80 | [47] |

Note: TP, true positive; FP, false positive; FN, false negative; TN, true negative; Ref, reference; NA, not available

**Table 4. Characteristics of studies on neutralising antibodies detection tests.**

| Author | Year | Study design | Reference test | Index test format | Index test (Commercial/ In-house) | Time of sample collection (day of post symptom onset) | Total number of samples | TP | FP | FN | TN | Ref |
|---|---|---|---|---|---|---|---|---|---|---|---|---|
| Goh | 2015 | Case-control | Indirect immunofluorescence antibody assay and haemagglutination inhibition | Epitope blocking ELISA | In-house | NA | 80 | 60 | 1 | 0 | 19 | [54] |
| Morey | 2010 | Cohort | RT-PCR and/or qRT-PCR or virus isolation | Peptide ELISA | In-house | NA | 28 | 17 | 2 | 2 | 7 | [55] |

Note: TP, true positive; FP, false positive; FN, false negative; TN, true negative; Ref, reference; NA, not available

There are three types of commercial tests: ELISA-based, IFA-based, and rapid test. The diagnostic performance of all the tests (ELISA and IFA) developed by Euroimmun (Lübeck, Germany) had more than 90% sensitivity and specificity. There was no heterogeneity found in the diagnostic performance of IFA (Table 9). Another ELISA-based assay was developed by Abcam (UK) and Inbios (Seattle, WA, USA). Both assays showed high diagnostic performance with no heterogeneity.

Meanwhile, ELISA-based and rapid IgM test developed by Standard Diagnostics Inc. (Yongin-si, South Korea) had poor diagnostic performance compared to tests of the same format from other manufacturers. The sensitivity of another IgM rapid test developed by CTK Biotech Inc. (San Diego, CA, USA) was equally poor (27.9%; CI 10.8 to 55.2). Compared to the sensitivity claimed by the manufacturers, the sensitivity of the two rapid tests reported in this study was relatively low. In summary, ELISA-based and IFA outperform rapid tests in terms of diagnostic performance among all the commercial tests.

## 3.8 Quality assessment

**3.8.1 Study design.** The diagnostic performance of the case-control and cohort/partial cohort partial case-control study design was compared (Table 10). No analysis was done for

**Table 5. Characteristics of the Index tests (n = 72) from the 35 included studies.**

| Characteristic | | No. (%) |
|---|---|---|
| Analyte | | |
| | IgM antibodies | 48 (66.7) |
| | IgG antibodies | 15 (20.8) |
| | Antibodies | 2 (2.8) |
| | Antigen | 7 (9.7) |
| Index test | | |
| | Commercial assay | 39 (54.2) |
| | In-house developed assay | 33 (45.8) |
| Index test format | | |
| | ELISA-based | 46 (63.9) |
| | Rapid test | 20 (27.8) |
| | Immunofluorescence assay | 6 (8.3) |
| Study design | | |
| | Cohort | 17 (23.6) |
| | Case-control | 18 (25) |
| | Partial cohort and partial case-control | 37 (51.4) |

**Fig 2. Forest plot for antigen detection test based on test format; CI, confidence interval; TP, true positive; FP, false positive; FN, false negative; TN, true negative.**

the antigen detection test since there is just one study with a case-control study design (Table 1). The sensitivity and specificity of the two study designs were shown to differ significantly ($P = <0.05$) for the IgM detection test. Meanwhile, only the specificity of the two study designs was shown to be significantly different for the IgG detection test. Overall, the case-control study had a higher diagnostic accuracy than the cohort/partial cohort partial case-control study.

**3.8.2 Risk of bias and application.**   Based on the QUADAS-2 tool, nine (24.3%) and six (16.2%) studies had a high risk of bias with regards to the patient selection and index test, respectively (Fig 8). All of the studies showed low applicability concern. The risk of bias and applicability concerns assessment of individual studies is available in S2 Fig.

## 3.9 Publication bias

Analysis showed a symmetrical funnel plot, suggesting no publication bias ($P = 0.236$) (Fig 9).

## 4. Discussion

CHIKV is a mosquito-borne virus that causes an acute febrile illness with severe joint pain. This study reviewed and analysed serological tests detecting CHIKV antigen, IgM, and IgG antibodies. During CHIKV infection, once the virus enters the host, it replicates and causes viremia, which lasts about 7 days. The patient's clinical manifestations, such as fever, are closely related to the high viral load during this period [56,57]. The appearance of the antibodies in the following phase is linked to a decrease in viremia. In this meta-analysis, the acute phase is defined as days 1 to 7 following the beginning of symptoms, while the convalescent phase is defined as after 7 days of symptom onset. Since different analytes are detected at

**Table 6. Analysis for antigen detection tests.**

| | Number of index test | Sample size | Pooled Sensitivity | | P-value | Pooled Specificity | | P-value |
|---|---|---|---|---|---|---|---|---|
| | | | Percentage [95% CI] | $I^2$ [95% CI] | | Percentage [95% CI] | $I^2$ [95% CI] | |
| Test format | | | | | | | | |
| Rapid test | 4 | 532 | 85.8 [65.6; 95.1] | 93.0% [85.2; 96.7] | 1[a] | 96.1 [81.9; 99.3] | 56.9% [0.0; 85.7] | 0.721[a] |
| ELISA-based | 3 | 348 | 82.2 [55.6; 94.4] | 95.1% [89.1; 97.8] | | 96.0 [89.9; 98.5] | 0.0% [0.0; 85.1] | |

Abbreviations: CI, confidence interval; ELISA, enzyme-linked immunosorbent assay; $I^2$, Inconsistency
[a] Mann-Whitney test

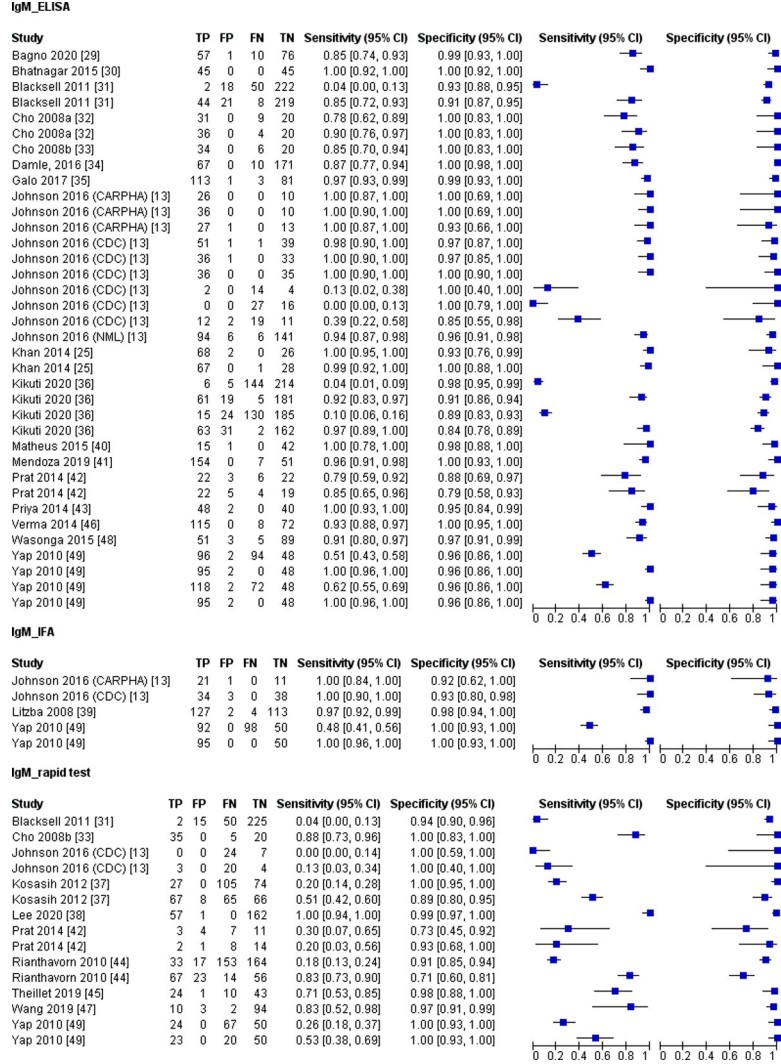

**Fig 3. Forest plot for IgM detection test based on test format; CI, confidence interval; TP, true positive; FP, false positive; FN, false negative; TN, true negative.**

different time points during CHIKV infection (acute and convalescent), herein, we elaborate the findings of this meta-analysis considering the utility of these tests during CHIKV infection.

## 4.1 Acute phase

Our meta-analysis demonstrates that antigen detection tests serve as a good diagnostic test for samples collected during the acute phase of CHIKV infection. According to the CHIKV testing algorithm developed by the Center for Disease Control and Prevention (CDC), qRT-PCR is the standard test used for samples collected less than 6 days after symptom onset [58]. Nevertheless, the qRT-PCR has limitations, such as the need for expensive reagents and equipment that are not available in most laboratories, especially in rural areas where CHIKV is prevalent. Less complicated tests, such as rapid and ELISA-based antigen detection tests, can be utilised as an alternative.

**Table 7. Analysis for IgM detection tests.**

| | Number of index test | Sample size | Pooled Sensitivity | | P-value | Pooled Specificity | | P-value |
|---|---|---|---|---|---|---|---|---|
| | | | Percentage [95% CI] | $I^2$ [95% CI] | | Percentage [95% CI] | $I^2$ [95% CI] | |
| Test format | | | | | | | | |
| ELISA-based | 31 | 5169 | 93.4 [81.7; 97.8] | 93.0% [91.3; 94.4] | 0.003 [a, b] | 96.8 [95.0; 98.0] | 37.4% [6.2; 58.2] | 0.796 [a] |
| Rapid test | 13 | 2040 | 42.3 [19.2; 69.4] | 92.2% [88.8; 94.6] | | 97.1 [92.0; 99.0] | 72.0% [52.9; 83.3] | |
| IFA | 4 | 739 | 99.3 [69.4; 100] | 91.0% [82.0; 95.5] | | 98.0 [93.6; 99.4] | 0.0% [0.0; 72.4] | |
| Commercial vs In-house | | | | | | | | |
| Commercial | 30 | 5388 | 78.6 [51.0; 92.8] | 94.0% [92.5; 95.1] | <0.001 [c] | 95.9 [93.3; 97.6] | 59.3% [41.2; 71.8] | 0.006 [c] |
| In-house | 18 | 2560 | 94.7 [87.7; 97.8] | 86.4% [80.4; 90.6] | | 98.0 [96.9; 98.8] | 0.0% [0.0; 0.0] | |
| Time of sample collection | | | | | | | | |
| ≤7 days | 10 | 2733 | 26.2 [9.0; 56.0] | 96.5% [95.0; 97.5] | <0.001 [c] | 95.8 [92.5; 97.7] | 52.4% [2.5; 76.8] | 0.914 [c] |
| >7 days | 12 | 1936 | 98.4 [90.7; 99.7] | 73.7% [53.3; 85.2] | | 96.6 [91.0; 98.8] | 69.9% [45.6; 83.4] | |

Abbreviations: CI, confidence interval; ELISA, enzyme-linked immunosorbent assay; IFA, Immunofluorescent assay; $I^2$, Inconsistency

[a] Kruskal-Wallis test

[b] pairwise tests ELISA-based vs rapid test, $P = 0.002$; pairwise test rapid test vs IFA, $P = 0.027$; pairwise test ELISA-based vs IFA, $P = 0.414$.

[c] Mann-Whitney test

For the antigen detection test, most of the studies in this meta-analysis employed samples from the early stage of infection (1 to 20 days). Virus isolation or a molecular-based assay were used as reference standards to confirm the presence of viral particles (antigen). Only one study [23] used the case-control study design, and all antigen tests were generated in-house. As a result, there was no further analysis based on these variables to discover the source of heterogeneity.

The low sensitivity of the test against different CHIKV genotypes could be one source of heterogeneity for antigen detection tests. The rapid test developed by Okabayashi et al. [26] was shown to be less sensitive in detecting CHIKV of Asian genotype [22]. Suzuki et al. [28] generated new monoclonal antibodies and showed that their improvised rapid test was more sensitive to cultured Asian and West African genotypes than the rapid test developed by Okabayashi et al. [26]. To further augment the diagnostic accuracy of this test, we suggest that different populations covering different genotypes should be tested in the future.

In this meta-analysis, the time of sample collection for IgM detection tests ranges from day 1 to day 40 post symptom onset. This wide range of time of samples collection is theoretically acceptable as IgM antibodies are known to appear as early as day 2 from the onset of illness and can persist up to 3 months [59]. However, our meta-analysis revealed that the sensitivity of the IgM detection tests was low for acute-phase samples (1 to 7 days post symptom onset) (26.2%) compared to the convalescent-phase samples (≥7 days post symptom onset) (98.4%). This result is consistent with Natrajan et al. (60) findings, who found that IgM tests can detect CHIKV with a 100% accuracy rate for samples taken more than 6 days of symptom onset. In summary, while IgM antibodies begin to develop from day 2 of CHIKV infection, the level can be way below the detection limit of most serological assays. Thus, the IgM detection test is not recommended for samples taken during the acute phase of infection.

## 4.2 Convalescent phase

As mentioned above, our meta-analysis showed that the diagnostic accuracy of the IgM detection test was high for convalescent-phase samples. According to WHO guidelines, a confirmed CHIKV case is defined as the presence of CHIKV IgM antibodies in a single serum sample taken during the acute or convalescent phases, indicating recent infection [10].

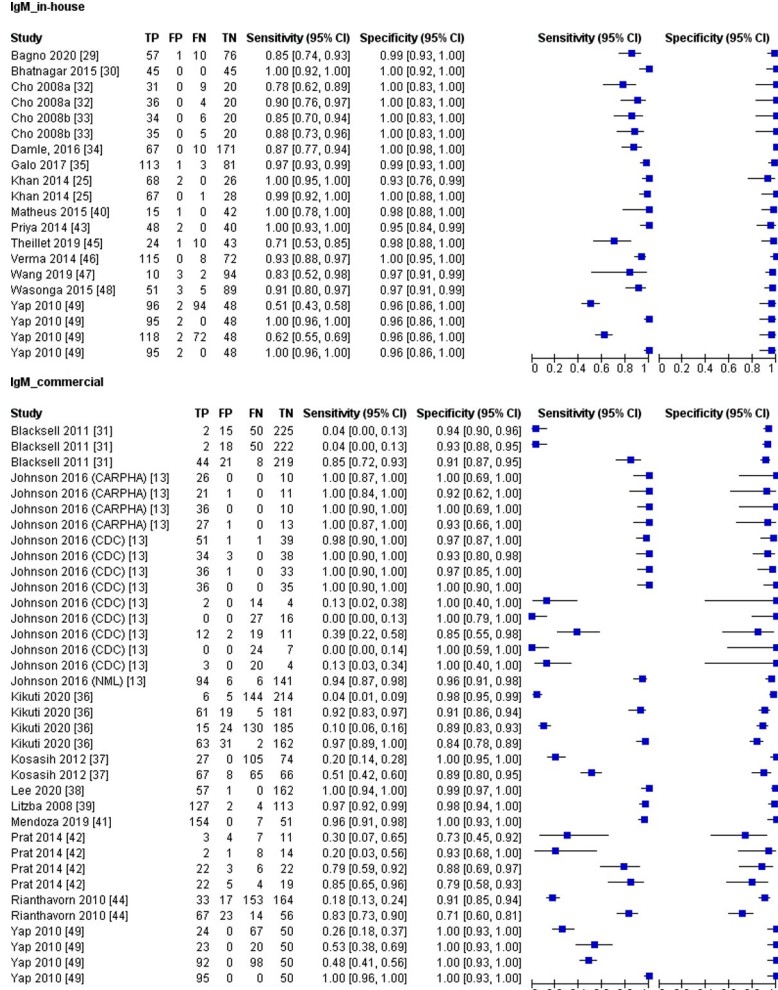

**Fig 4. Forest plot for IgM detection test based on in-house developed and commercial test; CI, confidence interval; TP, true positive; FP, false positive; FN, false negative; TN, true negative.**

IgM rapid tests had the lowest diagnostic performance compared to ELISA-based and IFA. Despite having the highest accuracy, IFA requires more expensive equipment and reagents. In addition, we found that in-house developed IgM tests had higher diagnostic performance compared to commercial tests. This finding is consistent with an external quality assurance report that found in-house developed ELISA tests to be more sensitive than commercial ELISA tests [60]. We are concerned that case-control design would lead to the overestimation of performance of the in-house developed IgM tests. Nonetheless, excluding the case-control studies from the meta-analysis showed a similar result (see S2 Table). Thus, this strengthens the notion that the accuracy of in-house developed tests is better than commercial tests.

According to the CDC testing algorithm, the PRNT is required to confirm a positive IgM test in diagnosing CHIKV disease [56]. Our meta-analysis showed that the IgM test had more than 97% specificity, regardless of test formats. More than half of the index tests evaluated in this meta-analysis included other pathogen positive samples (e.g., dengue, ONN, and RRV) in determining the cross-reactivity of the tests (partial cohort partial case-control study). These results validated the high specificity of the IgM tests, which could imply that PRNT may not be needed as a confirmatory test for positive cases determined by IgM tests.

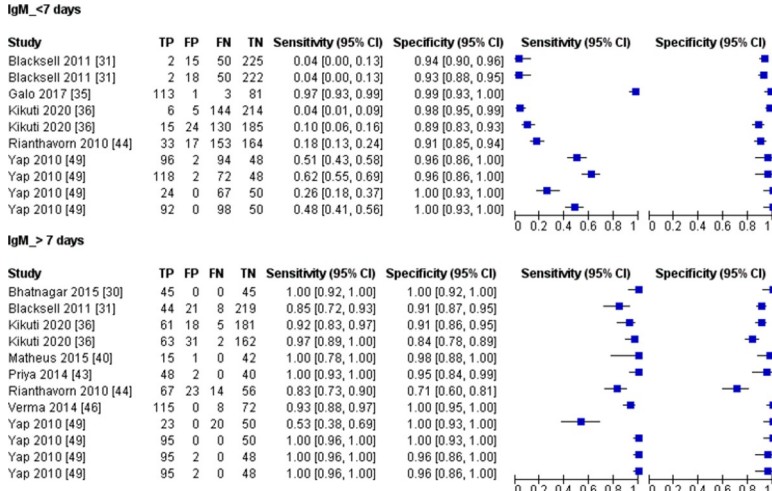

**Fig 5. Forest plot for IgM detection test based on time of sampling; CI, confidence interval; TP, true positive; FP, false positive; FN, false negative; TN, true negative.**

On the other hand, IgG antibodies can be detected approximately from day 7 to 10 post symptom onset and remain detectable for months to years [56]. Correspondingly, our meta-analysis showed that IgG detection tests had more than 93% sensitivity and specificity for samples collected between days 7 to 90 of post symptom onset. As CHIKV IgG antibodies persist for years, a second sample should be collected three weeks apart to rule out past infection. As stated in the WHO guidelines, a recent CHIKV diagnosis can be confirmed if there is a four-fold increase in IgG titer between the samples [10]. However, obtaining second samples from the patients is not always possible. In such situation, the presence of the CHIKV IgG antibody in a single sample should be interpreted in correlation with the clinical presentation of the patients.

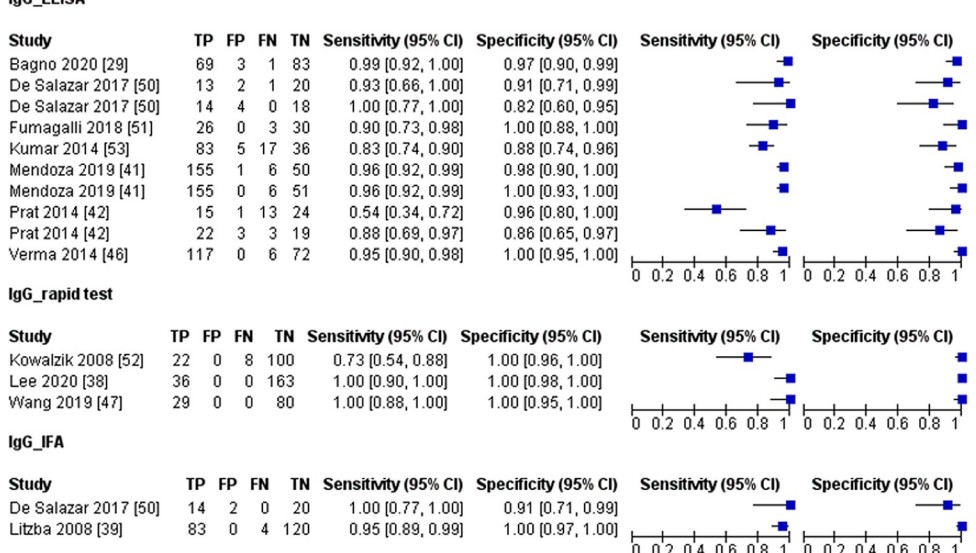

**Fig 6. Forest plot for IgG detection test based on test format; CI, confidence interval; TP, true positive; FP, false positive; FN, false negative; TN, true negative.**

**Table 8. Analysis for IgG detection tests.**

| | Number of index test | Sample size | Pooled Sensitivity | | P-value | Pooled Specificity | | P-value |
|---|---|---|---|---|---|---|---|---|
| | | | Percentage [95% CI] | $I^2$ [95% CI] | | Percentage [95% CI] | $I^2$ [95% CI] | |
| Test format | | | | | | | | |
| IFA | 2 | 243 | 96.0 [89.9; 98.5] | 0.0% [0.0; 0.0] | 0.269 [a] | 99.1 [61.0; 100] | 0.0% [0.0; 0.0] | 0.220 [a] |
| ELISA-based | 10 | 1147 | 93.0 [85.9; 96.6] | 83.6% [71.3; 90.6] | | 96.4 [91.2; 98.6] | 4.0% [0.0; 63.9] | |
| Rapid test | 3 | 438 | 99.3 [28.8; 100] | 0.0% [0.0; 0.0] | | 100 [0.0; 100] | 0.0% [0.0; 0.0] | |
| Commercial vs In-house | | | | | | | | |
| Commercial | 9 | 1038 | 95.3 [87.4; 98.4] | 82.3% [67.6; 90.3] | 0.475 [b] | 97.8 [91.6; 99.4] | 0.0% [0.0; 50.9] | 0.238[b] |
| In-house | 6 | 790 | 93.2 [82.8; 97.5] | 72.4% [36.3; 88.0] | | 99.6 [89.5; 100] | 0.0% [0.0; 59.9] | |

Abbreviations: CI, confidence interval; ELISA, enzyme-linked immunosorbent assay; IFA, Immunofluorescent assay; $I^2$, Inconsistency

[a] Kruskal-Wallis test

[b] Mann-Whitney test

There was no difference in the diagnostic performance of the IgG rapid test, IFA, and ELISA-based test. Among these tests, rapid tests are attractive because they are easy to perform, do not require expensive equipment, and the result can be obtained within a minute. Two commercial IgG rapid tests with promising diagnostic accuracy are recently available [38,47], but further evaluation with multiple prospective cohort studies is needed to provide comprehensive data for meta-analysis.

In summary, IgM and IgG antibody detection tests had high accuracy (>90%) for samples collected in the convalescent phase of CHIKV infection. The detection of IgM indicates recent infection, while a second sample collected at least 3 weeks apart is needed for the positive IgG test to rule out past infection.

## 4.3 Diagnostic performance of CHIKV commercial test kits

To our best knowledge, this is the first review that assessed the diagnostic accuracy of commercial tests for CHIKV. As mentioned above, the accuracy of the IgM detection test was very low for samples collected <7 days post symptom onset. Most commercial test kits recommend testing using samples collected between 6 to 8 days post symptom onset. Thus, we omitted acute-phase samples (< 7 days post symptom onset) from this analysis, and we found that the heterogeneity was low for almost all the commercial kits tested.

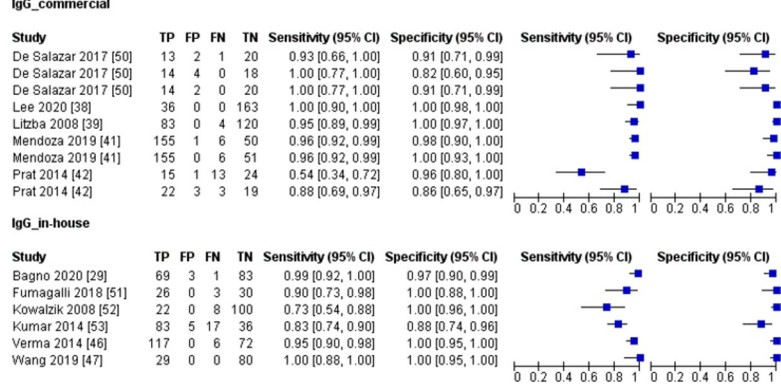

**Fig 7. Forest plot for IgG detection test based on in-house developed and commercial test; CI, confidence interval; TP, true positive; FP, false positive; FN, false negative; TN, true negative.**

**Table 9. Subgroup analysis for commercial tests.**

| | Manufacturer | Number of studies | Sample size | Pooled Sensitivity | | Sensitivity reported by manufacturer | Pooled Specificity | | Specificity reported by manufacturer |
|---|---|---|---|---|---|---|---|---|---|
| | | | | Percentage [95% CI] | $I^2$ [95% CI] | | Percentage [95% CI] | $I^2$ [95% CI] | |
| ELISA-based | | | | | | | | | |
| Anti-CHIKV ELISA (IgM) | Euroimmun Lübeck, Germany | 6 | 895 | 95.3 [92.9; 97.0] | 25.5% [0.0; 64.0] | 98.1 | 95.2 [84.9; 98.6] | 66.6% [20.3; 86.0] | 98.9 |
| Anti-CHIKV ELISA (IgG) | Euroimmun Lübeck, Germany | 3 | 295 | 95.5 [91.6; 97.6] | 30.4% [0.0; 92.8] | NA | 91.5 [78.0; 97.1] | 55.0% [0.0; 87.2] | NA |
| SD Chikungunya IgM ELISA | Standard Diagnostics Inc., Yongin-si, Korea | 2 | 336 | 65.3 [28.9; 89.8] | 93.9% [80.7; 98.1] | 93.6 | 90.9 [86.7; 93.9] | 0.0% [0.0; 0.0] | 95.9 |
| Anti-Chikungunya Virus IgM Human ELISA Kit | Abcam, UK | 2 | 116 | 100 [0; 100] | 0.0% [0.0; 0.0] | >90 | 97.7 [85.6; 99.7] | 0.0% [0.0; 0.0] | >90 |
| CHIKjj Detect MAC-ELISA | InBios, Seattle, WA, USA | 3 | 378 | 98.6 [64.9; 100] | 0.0% [0.0; 0.0] | >90 | 92.0 [87.9; 94.8] | 0.0% [0.0; 0.0] | >90 |
| Immunofluorescence assay (IFA) | | | | | | | | | |
| Anti-CHIKV IIFT (IgG) | Euroimmun Lübeck, Germany | 2 | 243 | 96.0 [89.9; 98.5] | 0.0% [0.0; 0.0] | 95 | 99.1 [61; 100] | 0.0% [0.0; 0.0] | 96 |
| Anti-CHIKV IIFT (IgM) | Euroimmun Lübeck, Germany | 4 | 499 | 98.1 [91.5; 99.6] | 0.0% [0.0; 0.0] | 100 | 98.6 [95.8; 99.5] | 0.0% [0.0; 72.6] | 96 |
| Rapid test | | | | | | | | | |
| On-site CHIK IgM Combo Rapid test | CTK Biotech Inc., San Diego, CA, USA | 3 | 145 | 27.9 [10.8; 55.2] | 81.0% [40.5; 93.9] | 90.4 | 98.7 [84.9; 99.9] | 0.0% [0.0; 0.0] | 98 |
| SD BIOLINE Chikungunya IgM | Standard Diagnostics Inc., Yongin-si, Korea | 3 | 216 | 19.1 [0.6; 90.0] | 80.7% [39.4; 93.8] | 97.1 | 73.3 [63.8; 81.0] | 0.0% [0.0; 0.0] | 98.9 |

Abbreviations: CI, confidence interval; NA, not available; $I^2$, Inconsistency.

Our meta-analysis supported the findings reported by Johnson et al. [13], which showed high diagnostic performance of the test kits manufactured by Euroimmun (Lübeck, Germany), Abcam (Cambridge, UK), and Inbios (Seattle, WA, USA). However, according to the authors, IFA developed by Euroimmun (Lübeck, Germany) needed more testing for equivocal results

**Table 10. Subgroup analysis for study design.**

| | Number of index test | Pooled Sensitivity | | P-value | Pooled Specificity | | P-value |
|---|---|---|---|---|---|---|---|
| | | Percentage [95% CI] | $I^2$ [95% CI] | | Percentage [95% CI] | $I^2$ [95% CI] | |
| IgM | | | | | | | |
| Case-control | 10 | 93.1 [86.3; 96.7] | 72.5% [47.9; 85.5] | 0.001 [a] | 99.3 [98.1; 99.7] | 0.0% [0.0; 0.0] | <0.001 [a] |
| Cohort/partial cohort partial case-control | 38 | 83.2 [62.2; 93.7] | 92.4% [90.5; 93.9] | | 96.1 [94.0; 97.5] | 47.4% [23.0; 64.0] | |
| IgG | | | | | | | |
| Case-control | 6 | 95.0 [89.6; 97.7] | 76.3% [46.8; 89.4] | <0.905 [a] | 99.8 [84.1; 100] | 0.0% | 0.015 [a] |
| Cohort/partial cohort partial case-control | 9 | 94.3 [82.6; 98.3] | 65.3% [29.4; 83.0] | | 94.6 [89.0; 97.4] | 0.0% [0.0; 58.1] | |

Abbreviations: CI, confidence interval; $I^2$, Inconsistency
[a] Mann-Whitney test

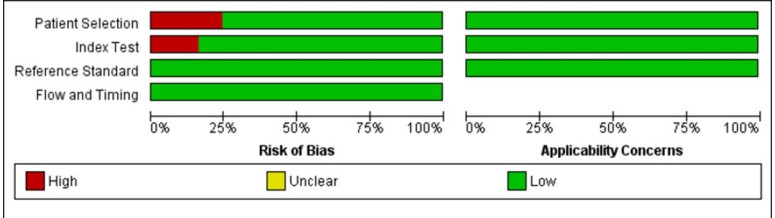

**Fig 8. Overall percentage of risk of bias and applicability concern using the QUADAS-2 tool.**

due to background fluorescence which may not be applicable in a real clinical setting. We also found that the diagnostic accuracy of most of the commercial tests reported in this review was lower than the accuracy mentioned by the manufacturers except for ELISA-based tests developed by Abcam (Cambridge, UK) and Inbios (Seattle, WA). Although the accuracy of these two tests was high, more studies using diverse samples population should be carried out to ascertain its use in other regions. Of note, among all the commercial tests evaluated in this review, only CHIKjj Detect MAC-ELISA (InBios, Seattle, WA, USA) has Conformité Européenne (CE) marking.

## 4.4 The impact of the study quality

**4.4.1 Study design.** The partial cohort partial case-control studies included other pathogens positive samples to evaluate the cross-reactivity of the tests. The ability of the tests to discern CHIKV from other pathogens is important because alphaviruses such as ONN and RRV are prevalent, especially in Sub-Saharan Africa and Australia. In tropical countries, samples positive for DENV are always used for specificity check due to the co-prevalence of CHIKV and DENV within the same region. The inclusion of these well-defined samples in partial

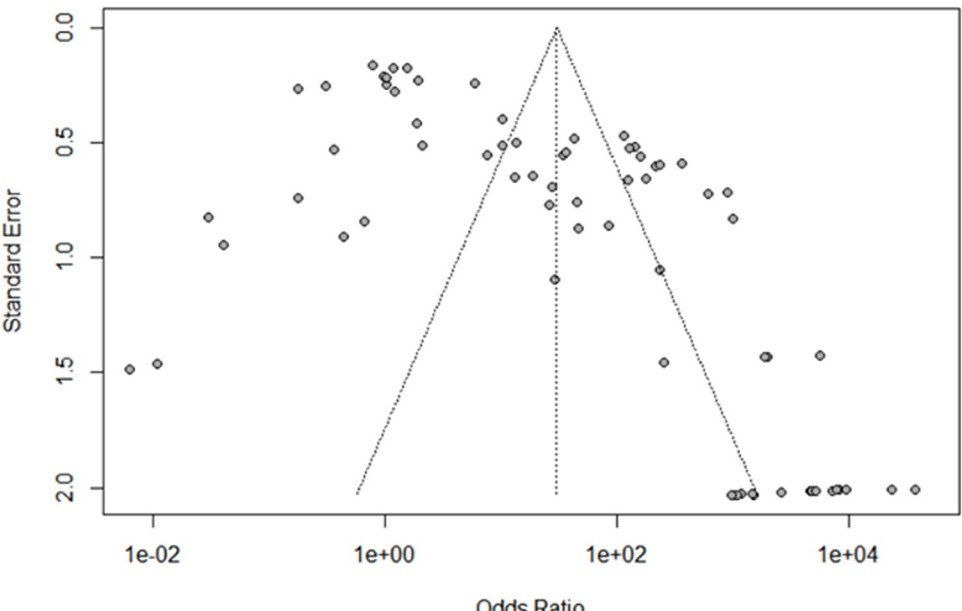

**Fig 9. Funnel plot asymmetry test to assess publication bias.** Each dot represents an individual study, and the dashed line represents the regression line. P-value = 0.236.

cohort partial case-control studies is unlikely to increase the risk of bias and thus were grouped with cohort studies.

One of the issues identified in most diagnostic accuracy studies is the flaw in the study design [61]. Because CHIKV patient samples are difficult to obtain, case-control studies are used for CHIKV diagnostic accuracy research. In this study design, the spectrum between individuals without chikungunya disease is widely separated from those with chikungunya disease. As a result, discerning between people who have the disease and those who have not is much easier. The case-control study design is expected to cause an overestimation in the diagnostic accuracy, which was observed in our analysis. The sensitivity and specificity of case-control study design were higher than cohort and partial cohort partial case-control study design, but not for the sensitivity of the IgG detection test. There was no statistical difference in the IgG sensitivity for the two study designs. Nevertheless, the case-control studies included samples from two distinct sources of populations (healthy and CHIKV positive), and these samples did not represent the population in a real clinical setting [62].

In summary, the cohort study design is the ideal study design in determining diagnostic accuracy. However, it is not always feasible for most studies, especially in countries with a low prevalence of chikungunya. Although the accuracy estimates from the case-control study design may not represent the actual value, this study design is an alternative to cohort study design, especially in determining the accuracy of a test in its developmental phase.

**4.4.2 Quality assessment of bias and application.**  The high risk of bias in the patient selection domain was mainly contributed by studies that applied case-control study design [23,30,38,39,41,45,46,52,54]. As mentioned previously, the case-control study design could exaggerate the test accuracy and thus may not reflect the actual accuracy.

For the index test domain, almost all the studies did not mention whether the index test results were interpreted without knowing the result of the reference standard. There is a high risk of bias as the interpretation of index test results can be influenced by knowledge of the reference standard. The ELISA-based test results were categorised into positive, borderline (or equivocal), and negative based on the obtained OD (absorbance) or a ratio. To simplify the analysis, some studies coded the borderline or equivocal samples as positive [41] and negative [13,50]. A study coded equivocal results for the immunochromatographic test as negative [31]. Although not described in the study, the equivocal result for ICT can be defined as ambiguous test lines observed. The inclusion of inconclusive results (borderline or equivocal) in the analysis will increase the risk of bias. However, as the number of borderline and equivocal samples (19 out of 10563) in this study was very small, the inclusion of these data will not affect the general result of the meta-analysis.

There was no bias recorded for the reference standard domain. Different reference standards were used in the diagnostic accuracy studies since no gold standard is available for diagnosing CHIKV. This meta-analysis specifies direct detection methods such as virus isolation and molecular-based method as the reference standard for antigen detection tests to ensure that the samples were collected during the viral stage. Some studies used molecular tests as the reference standards [36,41], and subsequent samples collected from the same patient were used for IgM or IgG accuracy studies. Although not wrong, each patient's immune response can be varied against CHIKV. Some patients may not develop antibodies against CHIKV, and thus the use of these samples could lead to the low sensitivity of the test. Analysis based on the reference standard used was not done in this study due to the variations of the reference standard even in a single study. Most studies briefly mentioned the reference standard used and did not provide detailed data. Thus, it is difficult to extract data based on the reported reference standard and perform further analysis.

For the flow and timing domain, the reference and index tests should be performed at the same or almost the same time point. However, as chikungunya disease is not available year-round, most of the studies in this review use retrospective samples (samples pre-defined in other studies) to determine the accuracy of the test, therefore we rated this domain as low risk of bias.

## 5. Strengths and limitations of the review

This systematic review and meta-analysis followed a standard protocol registered in the PROS-PERO database (CRD42021227523) and PRISMA-DTA review methodology. We carried out our literature search based on the quality of the study design, test formats, and type of analytes. We evaluated the diagnostic accuracy of serological tests detecting CHIKV antigen, IgM, and IgG antibody, which were applicable in different phases (acute and convalescent) of CHIKV infection. Furthermore, we analysed the diagnostic performance of the available commercial test kits for CHIKV and compared it with the diagnostic accuracy reported by the manufacturers.

Our review also has several limitations. Following the subgroup analysis, there was heterogeneity between groups that could not be explained by the findings of our study. This heterogeneity may be explained by performing analyses on other possible sources such as different lineages of CHIKV utilised to prepare antigen or antibody, the nature of antigen (recombinant protein or inactive virus), the sample population (country, origin, or CHIKV lineage), and types of the reference standards.

One of the characteristics that affect the width of the confidence interval is the sample size [63]. Due to the small sample size, some of the analyses showed a wide range of 95% confidence interval. For example, the heterogeneity ($I^2$) for the specificity of the rapid (95% CI 0 to 85.7) and ELISA-based (95%CI 0 to 85.1) antigen detection tests had a wide 95% confidence interval. A similar observation was seen for the specificity of the rapid IgG test (95% CI 0.0 to 100). Wide confidence often indicates that the estimated results provide less certain information. Therefore, at this point, we have a low level of certainty for analysis with a wide confidence interval.

In addition, only a subset of the studies included in this review provides information with respect to the time of sample collection. The sampling time is significant since the detection effectiveness of the test varies depending on the presence of analytes in the patient's sample. Other information such as blinding of the reference test result when interpreting index test, the expertise of the person who performs IFA, and samples conditions, are also important to grasp sources of variance and evaluate applicability. We strongly recommend employing a prospective cohort study design and a full report on the methodology associated with reference and index tests for a more accurate estimation of the diagnostic accuracy for CHIKV serological testing.

## 6. Conclusion

According to our meta-analysis, depending on the time of samples collection, antigen and antibody-based serological tests can accurately diagnose CHIKV. Antigen detection tests are an effective diagnostic test for samples obtained during the acute phase (1 to 7 days post symptom onset), whereas IgM and IgG detection tests can be used for samples collected in the convalescent phase (>7 days post symptom onset). In correlation to the clinical presentation of the patients, the combination of the IgM and IgG tests can differentiate recent and past infections. Several commercial IgM and IgG assays have been recognised as promising, which included kits from Euroimmun (Lübeck, Germany), Abcam (Cambridge, UK), and Inbios

(Seattle, WA). The caveats to the finding of this meta-analysis are inconclusive reporting of data in this review and low quality of reporting in diagnostic test accuracy studies.

## Supporting information

**S1 Checklist. PRISMA-DTA checklist.** Preferred reporting items for a systematic review and meta-analysis of diagnostic test accuracy studies.
(DOCX)

**S2 Checklist. PRISMA-DTA for the abstract checklist.**
(DOCX)

**S1 Appendix. Search strategy used for PubMed and CINAHL Complete and Scopus databases.**
(DOCX)

**S2 Appendix. QUADAS-2 Validation form.**
(DOCX)

**S1 Fig. Forest plot for antigen, IgM, IgG, and neutralising antibodies test; CI, confidence interval; TP, true positive; FP, false positive; FN, false negative; TN, true negative;** [**S1 Reference list**].
(TIF)

**S2 Fig. Risk of bias and applicability concerns assessment of individual studies using the QUADAS-2 tool;** [**S2 Reference list**].
(TIF)

**S1 Reference list. Reference list for S1 Fig.**
(PDF)

**S2 Reference list. Reference list for S2 Fig.**
(PDF)

**S1 Table.** Section A: Characteristics of commercial ELISA-based tests included in the meta-analysis. section B: Characteristics of commercial Immunofluorescence assays included in the meta-analysis. section C: Characteristics of commercial rapid tests included in the meta-analysis.
(DOCX)

**S2 Table. Analysis of commercial versus in-house developed IgM tests with the exclusion of case-control study.**
(DOCX)

## Acknowledgments

We acknowledge Dr. Mariska L. (University of Amsterdam) for providing valuable suggestions and Dr. Dayang E.Z. (Universiti Malaysia Sarawak) for her critical review of the articles.

## Author Contributions

**Conceptualization:** Anna Andrew, Ewe Seng Ch'ng, Thean-Hock Tang.

**Data curation:** Anna Andrew, Ernest Mangantig.

**Formal analysis:** Anna Andrew, Ernest Mangantig.

**Investigation:** Anna Andrew, Tholasi Nadhan Navien, Tzi Shien Yeoh, Ewe Seng Ch'ng.

**Methodology:** Anna Andrew, Ewe Seng Ch'ng, Thean-Hock Tang.

**Project administration:** Anna Andrew.

**Resources:** Anna Andrew.

**Software:** Anna Andrew, Ernest Mangantig.

**Supervision:** Thean-Hock Tang.

**Validation:** Ewe Seng Ch'ng.

**Visualization:** Marimuthu Citartan, Magdline S. H. Sum, Thean-Hock Tang.

**Writing – original draft:** Anna Andrew.

**Writing – review & editing:** Tholasi Nadhan Navien, Tzi Shien Yeoh, Marimuthu Citartan, Ernest Mangantig, Magdline S. H. Sum, Ewe Seng Ch'ng, Thean-Hock Tang.

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
