## [Decision Letter · Decision Letter 0]

14 Sep 2021

Dear Prof. Dr. Tang,

Thank you very much for submitting your manuscript "Diagnostic accuracy of serological tests for the diagnosis of Chikungunya virus infection: a systematic review and meta-analysis" for consideration at PLOS Neglected Tropical Diseases. As with all papers reviewed by the journal, your manuscript was reviewed by members of the editorial board and by several independent reviewers. In light of the reviews (below this email), we would like to invite the resubmission of a significantly-revised version that takes into account the reviewers' comments. 

Both reviewers raised important concerns about the study. I agree with them that the overall pooled sensitivity analysis is not worthwhile, as it is important to know (and compare) the performance of tests according to their characteristics and uses. Combining tests that detect IgM antibodies, IgG antibodies, neutralizing antibodies and antigen makes no sense as they have different uses and applications. Likewise, the combination of rapid tests (which show poor performance) with ELISA and IFA to obtain an overall sensitivity does not contribute to the understanding of the performance of these diagnostic methodologies. Just as important, it is not worth combining studies that evaluated samples from the acute and convalescent phases of the disease to determine the overall pooled sensitivity. 

I suggest that overall pooled analysis be removed and that the paper focus on the subgroup analysis, comparing test performance according to the type of test (RDT, ELISA, IFA), the type of antibody detected (IgM, IgG, …), and the timing of sampling (acute, convalescent). Such an approach will best answer which antibody (and which test method) should be used in the acute and in the convalescent-phases of the disease. If possible, further assessment of tests sensitivity according to study design and reference standard for comparison will be good.

Another concern is the fact that: “Most of the studies did not specify the time of sample collection”. The lack of this information introduces an important bias, making it difficult to properly interpret sensitivity because sampling time is critical to determine the performance of serological tests. As stated above, meta-analysis should be performed for subgroups based on the time of sampling (acute vs. convalescent). Please address these issues so that the manuscript can be considered for publication on PLOS NTD.

We cannot make any decision about publication until we have seen the revised manuscript and your response to the reviewers' comments. Your revised manuscript is also likely to be sent to reviewers for further evaluation.

Sincerely,

Guilherme S. Ribeiro, M.D., M.Sc., Ph.D

Associate Editor

Emma Wise

Deputy Editor

Both reviewers raised important concerns about the study. I agree with them that the overall pooled sensitivity analysis is not worthwhile, as it is important to know (and compare) the performance of tests according to their characteristics and uses. Combining tests that detect IgM antibodies, IgG antibodies, neutralizing antibodies and antigen makes no sense as they have different uses and applications. Likewise, the combination of rapid tests (which show poor performance) with ELISA and IFA to obtain an overall sensitivity does not contribute to the understanding of the performance of these diagnostic methodologies. Just as important, it is not worth combining studies that evaluated samples from the acute and convalescent phases of the disease to determine the overall pooled sensitivity. 

I suggest that overall pooled analysis be removed and that the paper focus on the subgroup analysis, comparing test performance according to the type of test (RDT, ELISA, IFA), the type of antibody detected (IgM, IgG, …), and the timing of sampling (acute, convalescent). Such an approach will best answer which antibody (and which test method) should be used in the acute and in the convalescent-phases of the disease. If possible, further assessment of tests sensitivity according to study design and reference standard for comparison will be good.

Another concern is the fact that: “Most of the studies did not specify the time of sample collection”. The lack of this information introduces an important bias, making it difficult to properly interpret sensitivity because sampling time is critical to determine the performance of serological tests. As stated above, meta-analysis should be performed for subgroups based on the time of sampling (acute vs. convalescent). Please address these issues so that the manuscript can be considered for publication on PLOS NTD.

Reviewer's Responses to Questions

**Key Review Criteria Required for Acceptance?**

**Methods**

-Are the objectives of the study clearly articulated with a clear testable hypothesis stated?

-Is the study design appropriate to address the stated objectives?

-Is the population clearly described and appropriate for the hypothesis being tested?

-Is the sample size sufficient to ensure adequate power to address the hypothesis being tested?

-Were correct statistical analysis used to support conclusions?

-Are there concerns about ethical or regulatory requirements being met?

Reviewer #1: Objectives are clearly stated, and methods are appropriate for an overall assessment of test accuracy. However, some methodological choices need to be better justified. For example, authors considered high risk of bias if a study excluded borderline or equivocal results from the analysis or used multiple reference tests. However, equivocal results are usually not to be interpreted as either positive or negative, thus assigning these cases as either TP/FP/FN/TN can biased the results. This is of particular importance if studies are assigning these cases differently. To that point, an overall description on how each study classified equivocal results will help understanding this potential source of heterogeneity. Similarly, for diseases in which no gold standard testing is adequate throughout most of the course of the disease, it is reasonable that a combination of tests is used to assure a correct classification. Authors also mention that only studies that used direct diagnosis methods (antigen detection, viral isolation or PCR) as reference standards were used. This limits the studied population to the cases that sought medical care early enough in the course of disease. Authors also opted to combine acute- and convalescent-phase data (when available) for an overall description of IgM and IgG testing accuracy. Although not wrong, these tests are not intended for acute-phase testing, thus the overall accuracy combining both phases has limited applicability.

Reviewer #2: This is a well written manuscript that is of particular interest to diagnosticians of tropical infections.

**Results**

-Does the analysis presented match the analysis plan?

-Are the results clearly and completely presented?

-Are the figures (Tables, Images) of sufficient quality for clarity?

Reviewer #1: Results are clearly presented although additional information might be useful to interpret the analysis. In Table 1, it would be important to know which clinical setting the studies took place (outpatient, hospitalized, etc). It would also be useful to know which type of reference test was used (antigen detection, viral isolation, PCR). The average and range of days of symptoms of the tested sample will also be important for interpretation of Table 3 summarizing IgM/IgG sensitivity. Similarly, in the subgroup analysis of commercial tests it might be also important to summarize the type of study and average days of symptom of samples used for these calculations (at a quick glance it seems like most are partial cohort/case-control for most companies and cohort for most SD). For Table 7, it is unclear what number of index tests mean. Is it the number of studies evaluating the test? Euroimmun only has one CHIKV ELISA IgM commercial test, for example, hence my confusion.

Reviewer #2: The results are well presented as are the figures.

**Conclusions**

-Are the conclusions supported by the data presented?

-Are the limitations of analysis clearly described?

-Do the authors discuss how these data can be helpful to advance our understanding of the topic under study?

-Is public health relevance addressed?

Reviewer #1: Conclusions are supported by the data presented. Some important limitations were not discussed. Authors state that most of the studies did not specify the time of sample collection. This can be an issue since the reference tests they selected for are all direct methods. If IgM/IgG testing is being done in the same sample as the reference test, this can artificially decrease sensitivity because it is being used at an inappropriate timing. This is likely one of the reasons for the heterogeneity found. Similarly, another source of heterogeneity not discussed it the intentional pooling of acute- and convalescent-phase samples for the overall sensitivity calculations, particularly for IgM/IgG. In the subgroup analysis, there is likely a combination of factors that were only assessed individually that potentially justifies the heterogeneity (e.g. acute- vs convalescent-phase samples, study type, etc.). Additionally, although generally the statement “Detection of IgG alone indicates past infection, while IgM positive with or without IgG indicates recent infection” is correct, I would point to caution as detection of IgG alone might not necessarily indicate past infection if a false negative in the IgM testing occurs.

Reviewer #2: The authors have missed some of the main limitations especially sample timing, reference comparators and sample composition. They are mentioned but only in passing - more emphasis needs to be placed on these important potential bias factors. 

I am concerned about the lack of detail in the timing of the sample (days of illness) information provided in this meta-analysis. While the authors state that this was a problem in the abstract it is not mentioned as a limitation of the study. It is noted that this information is included in table 5 of the study and illustrates the point well regarding sample timing especially in the acute phase of the study. The authors have overlooked the importance of this observation as it is imperative that the days of illness be included in any diagnostic accuracy study because if there is a large number of acute phase samples included in the study then it is unlikely that there will be detectable levels of antibodies available and therefore this will be reflected in the results indicating low sensitivity as illustrated in the last rows of table 5. This conclusion is fine because you cannot detect something that is not present. Conversely, it is simple to make any insensitive test look good by including large numbers of convalescent samples in the study group. It is strongly recommended that additional comment or information be provided in the Discussion section regarding the impact of the timing of the sample in the context of days of illness presentation.

There is little or no mention of the influence of Chikungunya prevalence on the outcome of diagnostic accuracy studies. While there is mention of this in the section on bias the implications are not really clear to the reader (i.e., high vs low prevalence in prospective study design and high versus low n positives in a case-control design) Please provide additional information regarding the implications of such bias in the sample composition especially in case control studies.

There is an absence of information regarding the influence of the reference comparator in the diagnostic accuracy study. This is an especially important source of bias often an inappropriate reference comparator is selected or an inappropriate diagnostic cutoff is used which can significantly bias the results. Furthermore, it is often more appropriate that a final patient case results (true disease status) of “Chikungunya positive” or “Chikungunya negative” be used for calculating the sens and spec because the true disease status is more relevant that comparing positivity scores against an randomly assigned diagnostic cutoff. 

Line 420. The authors have stated the following “ Second, we received no response from the corresponding authors (24, 32) regarding discrepancies in the raw data and diagnostic accuracy.” It is not clear what the authors mean by this statement as there is no additional information in the manuscript regarding these two papers. It would appear that because these two studies departed from the main theme of generally high diagnostic accuracy that there is a methodological problem with the study. The study of Huits et al describes the Arkay Chikungunya antigen ICT which described “fair diagnostic sensitivity for ECSA genotype chikungunya, but low sensitivity for Asian genotype, and poor overall specificity.”The study of Blacksell et al describes a diagnostic accuracy study evaluating the Standard Diagnostics IgM ELISA using a series of acute-phase samples compared to a gold standard reference comparators with the conclusion that the essay had low sensitivity which is a reflection on the days of illness of the samples. In fact, figure 4, indicates that for that particular study there was Low risk of bias or applicability concerns. Because of the lack of detail, I recommend that this sentence be removed from the manuscript as the implications are unfair to the authors.

**Editorial and Data Presentation Modifications?**

Reviewer #1: L87: The number of CHIKV serological tests increase tremendously after the Indian Ocean outbreaks in 2004

- As in number of tests used or different tests commercially available?

Line 126: We included only the optimized index test data in the analysis for studies developing serological tests using a different batch of antigens or antibodies.

- What is optimized index test data?

Line 169: For commercial tests (specific brand) with two or more reports, meta-analyses were done to determine their diagnostic accuracy. We included only samples collected after 7 days from the onset of clinical symptoms for this analysis.

- Clarify that for 2+ studies describing accuracy of a commercially available test, a meta-analysis per manufacturer/brand was done. This sentence was not clear to me until I read the results.

Clarification on the use of “acceptable reference standards” (line 347) is needed. Are the authors suggesting the reference standards they used (PCR, viral isolation or antigen detection) are not acceptable?

Reviewer #2: (No Response)

**Summary and General Comments**

Reviewer #1: The overall accuracy analysis seems to have limited applicability due to the compilation of studies with such different methodologies (e.g., grouping different detection methods – ELISA, RDT, IF; sample type, etc.). In that sense, the subgroup analysis is more informative and perhaps the most important aspect of this work. The risk of bias assessment needs better theoretical background justifying authors choices in qualifying the information (i.e., why were studies that used a combination of reference tests classified as having higher risk of bias?). Perhaps the novelty factor of a meta-analysis lies on the discussion of the heterogeneity of the data. As described, many aspects of this discussion were lacking.

Reviewer #2: The authors have missed some of the main limitations especially sample timing, reference comparators and sample composition. They are mentioned but only in passing - more emphasis needs to be placed on these important potential bias factors to provide a balanced viewpoint.

PLOS authors have the option to publish the peer review history of their article (what does this mean?). If published, this will include your full peer review and any attached files.

Reviewer #1: No

Reviewer #2: No
---

## [Decision Letter · Decision Letter 1]

3 Dec 2021

Dear Prof. Dr. Tang,

Thank you very much for submitting your manuscript "Diagnostic accuracy of serological tests for the diagnosis of Chikungunya virus infection: a systematic review and meta-analysis" for consideration at PLOS Neglected Tropical Diseases. As with all papers reviewed by the journal, your manuscript was reviewed by members of the editorial board and by several independent reviewers. The reviewers appreciated the attention to an important topic. Based on the reviews, we are likely to accept this manuscript for publication, providing that you modify the manuscript according to the review recommendations. 

Sincerely,

Guilherme S. Ribeiro, M.D., M.Sc., Ph.D

Associate Editor

Emma Wise

Deputy Editor

Reviewer's Responses to Questions

**Key Review Criteria Required for Acceptance?**

**Methods**

-Are the objectives of the study clearly articulated with a clear testable hypothesis stated?

-Is the study design appropriate to address the stated objectives?

-Is the population clearly described and appropriate for the hypothesis being tested?

-Is the sample size sufficient to ensure adequate power to address the hypothesis being tested?

-Were correct statistical analysis used to support conclusions?

-Are there concerns about ethical or regulatory requirements being met?

Reviewer #1: Objectives are clearly stated, and methods are appropriate. I appreciate that the authors justified some of their methodological choices so the readers can easily understand the rationale some assessments are based. Authors also justified that there was a low proportion of equivocal results, so the potential impact in the analysis is minimal (although there was not a clear statement of what that proportion was).

Reviewer #2: The revisions made by the authors are acceptable.

**Results**

-Does the analysis presented match the analysis plan?

-Are the results clearly and completely presented?

-Are the figures (Tables, Images) of sufficient quality for clarity?

Reviewer #1: Results are clearly presented although some figures might be combined for a more concise display of the information. It would still be useful to have a description of both the clinical setting the studies took place (outpatient, hospitalized, etc) and geographic location of where the studies took place to have an idea of the epidemiological scenario in which it took place. A brief text summary (e.g., most (XX%) of studies occurred with outpatient patients retrospectively and XX% took place in endemic regions) would suffice. That gives the reader context to what these studies represent.

Reviewer #2: The revisions made by the authors are acceptable.

**Conclusions**

-Are the conclusions supported by the data presented?

-Are the limitations of analysis clearly described?

-Do the authors discuss how these data can be helpful to advance our understanding of the topic under study?

-Is public health relevance addressed?

Reviewer #1: Conclusions are supported by the data presented although some phrasing needs to be revised. Some limitations were discussed as the results were presented (IgG RDT having higher accuracy) but need to be reinforced in the discussion as well. Additionally, although generally the statement “Detection of IgG alone indicates past infection, while IgM positive with or without IgG indicates recent infection” is correct, I would point to caution as detection of IgG alone might not necessarily indicate past infection if a false negative in the IgM testing occurs. Lastly, the authors mention throughout the discussion/conclusion that IgM and IgG testing of the convalescent-phase sample is recommended to differentiate between recent and past infections. I think the authors might mean testing IgM/IgG in the acute-phase sample to help differentiate between recent and past infections, please revise. Although IgM sensitivity is low for acute-phase samples, testing only convalescent-phase samples would not differentiate between recent or past infection since the patient would have had enough time to develop antibodies even in a primary infection.

Reviewer #2: The revisions made by the authors are acceptable.

**Editorial and Data Presentation Modifications?**

Reviewer #1: Figs 2-7 seems to be somewhat redundant with Figure S1.

- The stratification is relevant, but in interest of efficiency I would suggest combining them into maybe Figure S1 with three additional columns to denote the test type (RDT, ELISA), whether or not it is a commercial test, and days of symptoms (> or < 7)

Ag detection test: no difference between rapid test and elisa

- Very large confidence for Sp heterogeneity, I would like to have the authors discuss possible explanations in the discussion

- Include I² in the legend for Table 6

Line 251: In addition, the sensitivity of the rapid tests was statistically different from ELISA-based (93.4%; 95% CI 81.7 to 97.8; P=0.002) and IFA (99.3%; 95% CI 69.4 to 100; P=0.027)

- I would suggest adding RDTs Se in the phrasing as well for easy reading “In addition, the sensitivity of the rapid tests (42.3%, 95%CI…) …”

Line 267: The forest plot (Fig 5) shows that the sensitivity for samples collected ≤7 days of symptoms onset mostly lies on the left side of the plot. Incoherence with this observation, our meta-analysis shows that the sensitivity of the acute samples was significantly lower than convalescent samples (Table 7).

- It doesn’t seem incoherent. Fig 5 shows most sensitivities within the ≤7 days studies around 0-60% (left skewed) and most sensitivities within >7 days studies around 80-100% (right skewed), meaning lower sensitivity in acute-phase samples.

Table S1: Clarify if CHIKjj Detect MAC-ELISA is IgM or IgG (both are available commercially), same for Anti-Chikungunya Virus, Abcam. Similarly for Table 9.

Figure 9 could be submitted as a supplementary figure

Line 426: “As IgG antibodies generally persist for years, qualitative detection of both CHIKV IgG and IgM antibodies in the convalescent phase can help to differentiate recent and past infections.”

- I think the authors might mean testing IgM/IgG in the acute-phase sample to help differentiate between recent and past infections, please revise. Although IgM sensitivity is low for acute-phase samples, testing only convalescent-phase samples would not differentiate between recent or past infection since the patient would have had enough time to develop antibodies even in a primary infection.

- Same comment for line 435: “In summary, IgM and IgG antibody detection tests need to be carried out simultaneously for a more accurate diagnosis of CHIKV infection in the convalescent phase of the infection.”

- And Line 537: “In contrast, IgM and IgG tests can be used for samples collected in the convalescent phase, whereby the combination of the tests can differentiate recent and past infections.”

Line 429: “Rapid tests showed the highest diagnostic accuracy among test formats available to detect

CHIKV IgG antibodies, followed by IFA and ELISA-based tests”

- A follow up phrase pointing that there was no statistical difference between RDT, ELISA and IFA sensitivities and that the confidence intervals for the RDT Se and Sp are very large (Table 8) is needed here.

Reviewer #2: The revisions made by the authors are acceptable however the manuscript still requires some editing for English grammar and phrasing.

**Summary and General Comments**

Reviewer #1: The manuscript improved significantly with changes in the methodology and text and my main concerns were addressed. Few revisions are still necessary to assure some phrasings are not misinterpreted.

Reviewer #2: The revisions made by the authors are acceptable.

PLOS authors have the option to publish the peer review history of their article (what does this mean?). If published, this will include your full peer review and any attached files.

Reviewer #1: No

Reviewer #2: No

Figure Files:

Data Requirements:

Reproducibility:

References

---

## [Editor Report · Decision Letter 2]

6 Jan 2022

Dear Prof. Dr. Tang,

We are pleased to inform you that your manuscript 'Diagnostic accuracy of serological tests for the diagnosis of Chikungunya virus infection: a systematic review and meta-analysis' has been provisionally accepted for publication in PLOS Neglected Tropical Diseases.

Best regards,

Guilherme S. Ribeiro, M.D., M.Sc., Ph.D

Associate Editor

Emma Wise

Deputy Editor

---

## [Editor Report · Acceptance letter]

21 Jan 2022

Dear Prof. Dr. Tang,

We are delighted to inform you that your manuscript, "Diagnostic accuracy of serological tests for the diagnosis of Chikungunya virus infection: a systematic review and meta-analysis," has been formally accepted for publication in PLOS Neglected Tropical Diseases.

Best regards,

Shaden Kamhawi

co-Editor-in-Chief

Paul Brindley

co-Editor-in-Chief
